# Cardiovascular health metrics from mid- to late-life and risk of dementia: A population-based cohort study in Finland

Yajun Liang[1,2], Tiia Ngandu[3,4], Tiina Laatikainen[3,5,6], Hilkka Soininen[7,8], Jaakko Tuomilehto[3,9], Miia Kivipelto[4,8,10,11]*, Chengxuan Qiu[1]*

**1** Aging Research Center & Center for Alzheimer Research, Department of Neurobiology, Care Sciences and Society (NVS), Karolinska Institutet and Stockholm University, Stockholm, Sweden, **2** Department of Global Public Health, Karolinska Institutet, Stockholm, Sweden, **3** Public Health Promotion Unit, Finnish Institute for Health and Welfare, Helsinki, Finland, **4** Division of Clinical Geriatrics & Center for Alzheimer Research, NVS, Karolinska Institutet, Stockholm, Sweden, **5** Institute of Public Health and Clinical Nutrition, University of Eastern Finland, Kuopio, Finland, **6** Joint Municipal Authority for North Karelia Social and Health Services (Siun Sote), Joensuu, Finland, **7** Neurocenter, Department of Neurology, Kuopio University Hospital, Kuopio, Finland, **8** Department of Neurology, Institute of Clinical Medicine, University of Eastern Finland, Kuopio, Finland, **9** Saudi Diabetes Research Group, King Abdulaziz University, Jeddah, Saudi Arabia and Department of Public Health, University of Helsinki, Helsinki, Finland, **10** Neuroepidemiology and Ageing Research Unit, School of Public Health, Imperial College London, London, United Kingdom, **11** Stockholms Sjukhem, Research and Development Unit, Stockholm, Sweden

* miia.kivipelto@ki.se (MK); chengxuan.qiu@ki.se (CQ)

**Data Availability Statement:** Data are from the CAIDE project, a population-based cohort study on the associations of social, lifestyle, and cardiovascular risk factors from midlife onwards

## Abstract

### Background

Very few studies have explored the patterns of cardiovascular health (CVH) metrics in mid-life and late life in relation to risk of dementia. We examined the associations of composite CVH metrics from midlife to late life with risk of incident dementia.

### Methods and findings

This cohort study included 1,449 participants from the Finnish Cardiovascular Risk Factors, Aging, and Dementia (CAIDE) study, who were followed from midlife (baseline from1972 to 1987; mean age 50.4 years; 62.1% female) to late life (1998), and then 744 dementia-free survivors were followed further into late life (2005 to 2008). We defined and scored global CVH metrics based on 6 of the 7 components (i.e., smoking, physical activity, and body mass index [BMI] as behavioral CVH metrics; fasting plasma glucose, total cholesterol, and blood pressure as biological CVH metrics) following the modified American Heart Association (AHA)'s recommendations. Then, the composite global, behavioral, and biological CVH metrics were categorized into poor, intermediate, and ideal levels. Dementia was diagnosed following the Diagnostic and Statistical Manual of Mental Disorders, Fourth Edition (DSM-IV) criteria. Data were analyzed with Cox proportional hazards and the Fine and Gray competing risk regression models. During the follow-up examinations, dementia was diagnosed in 61 persons in 1998 and additional 47 persons in 2005 to 2008. The fully adjusted hazard ratio (HR) of dementia was 0.71 (95% confidence interval [CI]: 0.43, 1.16; $p$ = 0.174) and

with late-life cognitive phenotypes (http://www.uef.fi/caide/). Access to these original data is available to the research community upon approval by the CAIDE data management and maintenance committee. Applications for accessing these data can be submitted to Alina Solomon (alina.solomon@ki.se) at Karolinska Institutet, Sweden.

**Funding:** This work was supported in part by grants from the Academy of Finland (number: 278457, recipient: MK; number: 305810, recipient: MK; number: 317465, recipient: MK); the Academy of Finland (number: 334804, recipient: TN) and the Swedish Research Council (number: 2019-02226, recipient: MK) for the EU Joint Program on Neurodegenerative Diseases (JPND) project EURO-FINGERS under the aegis of JPND; the Academy of Finland (number: 291803, recipient: HS) for MIND-AD project; the Swedish Research council for Health, Working Life and Welfare (number: NA, recipient: MK); the Finnish Cultural Foundation (number: NA, recipient: TN); the Juho Vainio Foundation (number: NA, recipient: TN); the Jalmari and Rauha Ahokas Foundation, Finland (number: NA, recipient: TN); Alzheimerfonden Sweden (number: AF556161, recipient: MK); the Alzheimer's Research and Prevention Foundation (number: NA, recipient: MK), the Center for Innovative Medicine (CIMED) at Karolinska Institutet South Campus (number: NA, recipient: MK); the AXA Research Fund (number: NA, recipient: MK), the Knut and Alice Wallenberg Foundation (number: NA, recipient: MK); Stiftelsen Stockholms sjukhem (number: NA, recipient: MK), Konung Gustaf V:s och Drottning Victorias Frimurarstiftelse (number: NA, recipient: MK), and Hjärnfonden (number: 2015-0217, recipient: MK); the EU Seventh Framework Programme (HATICE) (number: 305374, recipient: MK); and US Alzheimer's Association (number: NA, recipient: MK). In addition, the work was supported by grants from the Swedish Research Council (number: 2015-02531, recipient: CQ; number: 2017-00740, recipient: CQ; number: 2017-05819, recipient: CQ); the Swedish Foundation for International Cooperation in Research and Higher Education for the Joint China-Sweden Mobility program (number: CH2019-8320, recipient: CQ); and Karolinska Institutet (number: 2018-01854, recipient: CQ; number: 2020-01456, recipient: CQ), Stockholm, Sweden. The funders had no role in study design, data collection and analysis, decision to publish, or preparation of the manuscript.

**Competing interests:** The authors have declared that no competing interests exist.

**Abbreviations:** AHA, American Heart Association; APOE, apolipoprotein E; ARIC, Atherosclerosis Risk

0.52 (0.29, 0.93; $p = 0.027$) for midlife intermediate and ideal levels (versus poor level) of global CVH metrics, respectively; the corresponding figures for late-life global CVH metrics were 0.60 (0.22, 1.69; $p = 0.338$) and 0.91 (0.34, 2.41; $p = 0.850$). Compared with poor global CVH metrics in both midlife and late life, the fully adjusted HR of dementia was 0.25 (95% CI: 0.08, 0.86; $p = 0.028$) for people with intermediate global CVH metrics in both midlife and late life and 0.14 (0.02, 0.76; $p = 0.024$) for those with midlife ideal and late-life intermediate global CVH metrics. Having an intermediate or ideal level of behavioral CVH in both midlife and late life (versus poor level in both midlife and late life) was significantly associated with a lower dementia risk (HR range: 0.03 to 0.26; $p < 0.05$), whereas people with midlife intermediate and late-life ideal biological CVH metrics had a significantly increased risk of dementia ($p = 0.031$). Major limitations of this study include the lack of data on diet and midlife plasma glucose, high rate of attrition, as well as the limited power for certain subgroup analyses.

## Conclusions

In this study, we observed that having the ideal CVH metrics, and ideal behavioral CVH metrics in particular, from midlife onwards is associated with a reduced risk of dementia as compared with people having poor CVH metrics. Maintaining life-long health behaviors may be crucial to reduce late-life risk of dementia.

## Author summary

### Why was this study done?

- Dementia is a global public health problem, but there is currently no cure or a disease-modifying therapy for dementia.

- Simulation studies suggested that interventions targeting modifiable risk factors (e.g., cardiovascular factors) could prevent up to one-third of dementia cases.

- A better understanding of the life-long cardiovascular health (CVH) metrics and risk of dementia may facilitate the development of optimal intervention strategies.

### What did the researchers do and find?

- We examined the associations of CVH metrics in midlife and late life with risk of incident dementia in a population-based cohort of 1,449 participants in Finland followed for around 30 years.

- Compared with poor CVH metrics, the ideal global and behavioral CVH metrics in midlife were associated with a reduced risk of dementia, whereas the ideal biological CVH metrics in late life appeared to be associated with an increased risk of dementia.

- Having an intermediate or ideal level of behavioral CVH metrics from midlife onwards was associated with a late-life reduced risk of dementia.

in Communities; BMI, body mass index; CAIDE, Cardiovascular Risk Factors, Aging, and Dementia; CI, confidence interval; CVH, cardiovascular health; DBP, diastolic blood pressure; DSM-IV, Diagnostic and Statistical Manual of Mental Disorders, Fourth Edition; HR, hazard ratio; MMSE, Mini-Mental State Examination; SBP, systolic blood pressure; STROBE, Strengthening the Reporting of Observational Studies in Epidemiology.

## What do these findings mean?

- The association of ideal global CVH metrics with a reduced dementia risk disappeared from midlife to old age, driven largely by the age-varying association between biological CVH metrics and risk of dementia.

- Maintaining a life-long optimal level of CVH metrics, especially behavioral health metrics, may reduce late-life risk of dementia.

- The association of late-life ideal biological CVH metrics with an increased risk of dementia may largely reflect the potential of reverse causality.

## Introduction

Dementia has been recognized as a global public health priority owing to its tremendous economic and societal burden [1]. Despite massive global efforts in the past 4 decades, no cure or even a disease-modifying therapy has been developed for dementia. Encouragingly, epidemiological studies have identified a range of life-long modifiable risk factors for dementia; of them, cardiovascular risk factors from midlife onwards have been shown to play a pivotal role in the development and onset of dementia [2]. Furthermore, simulation research has estimated that up to one-third of global dementia cases might be attributable to these modifiable risk factors [3]. Indeed, numerous epidemiological studies have reported that major cardiovascular risk factors (e.g., smoking and diabetes) and cardiovascular disease (e.g., heart failure and atrial fibrillation) are associated with an elevated risk of dementia [2,4–7]. The associations of cardiometabolic risk factors (e.g., high blood pressure, obesity, and high cholesterol) with the risk of dementia may vary with age from midlife to later in life [8–10]. However, very few studies have explored the potential associations of optimal cardiovascular health (CVH) metrics from mid- to late life with the risk of dementia. This is of high relevance from the public health perspective.

In 2010, the American Heart Association (AHA) defined ideal CVH based on 7 risk factors, also known as Life's Simple 7, to promote CVH [11]. These CVH metrics include 4 behavioral metrics (e.g., smoking, physical activity, diet, and body weight) and 3 biological metrics (e.g., blood glucose, blood cholesterol, and blood pressure) [11]. The CVH metrics have been shown to be a useful tool for predicting cardiovascular events because numerous studies have linked ideal CVH metrics with a lower risk of coronary heart disease and stroke [12,13].

The AHA's CVH metrics approach may help achieve healthy brain aging [14]. Having the ideal CVH metrics in midlife has been associated with a lower risk of dementia in late life [15,16]. However, population-based studies investigating the association between late-life composite CVH metrics and risk of dementia have yielded mixed results; some studies suggested an association between a higher CVH metric score in late life and a reduced risk of dementia [17,18], whereas others showed no association [19]. However, most of the previous studies have examined the CVH metrics assessed in either midlife or later in life. The association between changes or patterns of CVH metrics from midlife to late life and risk of dementia have been rarely examined in prospective studies. In addition, very few studies that examine the composite CVH metrics in midlife and late life in association with dementia risk have differentiated behavioral and biological CVH metrics [16]. This is important because the associations between certain biological components of CVH metrics and risk of dementia vary with age from middle age to late life [8–10].

Using data from the Cardiovascular Risk Factors, Aging and Dementia (CAIDE) in Finland, we sought to examine the association between CVH metrics from midlife to late life and risk of dementia. We hypothesized that optimal CVH metrics, especially occurring from midlife onwards, are associated with a reduced risk of incident dementia.

## Materials and methods

### Study design and participants

The study participants were derived from the Finnish CAIDE study, which is an ongoing population-based cohort study. The study design and purpose of CAIDE study were fully described elsewhere [20–22]. Briefly, the CAIDE study aims to investigate the associations of social, lifestyle, and cardiovascular risk factors from midlife onwards with late-life cognitive phenotypes. The initial participants of the CAIDE study were derived from population-based random samples examined within the framework of the North Karelia project [23] and the FINMONICA studies in 1972 to 1987 [24].

Fig 1 shows the flowchart of participants in the midlife and late-life examinations for this study. Briefly, of the eligible participants (n = 2,293) who were alive by the end of 1997 and were living in 2 geographically defined areas in or close to the towns of Kuopio and Joensuu (target population) [21], a random sample of 2,000 persons was selected for the first late-life evaluation in 1998. Of these, 1,449 (72.5%) undertook the examination in 1998, which consisted of the analytical sample for the association between midlife CVH metrics and late-life dementia detected in the 1998 examinations and the later wave of examination (analytical sample 1 in Fig 1).

The second late-life examination was conducted in 2005 to 2008. A total of 1,426 persons out of the initial 2,000 persons were still alive in the beginning of 2005 and eligible for the second late-life examination. Of these, 517 persons were not included due to death (n = 405, 78.3%) or poor health or refusal (n = 112, 21.7%), and 909 persons undertook the second late-life examination in 2005 to 2008. Of these, 165 persons either had dementia in 1998 or had missing information on dementia diagnosis, thus, 744 (81.8%) persons, who were free of dementia in 1998 and undertook the 2005 to 2008 examination, were included in the analyses involving late-life CVH metrics and dementia (analytical sample 2 in Fig 1).

The average follow-up time was 21.2 years from baseline to the 1998 examination and 8.3 years from the 1998 examination to the examination in 2005 to 2008.

### Ethics statement

The CAIDE study received approval from the local ethics committee at Kuopio University and the Kuopio University Hospital in Kuopio, Finland as well as from the ethics committee at Karolinska Institutet in Stockholm, Sweden. The verbal informed consent (midlife surveys in 1972 to 1987) or the written informed consent (late-life surveys in 1998 and 2005 to 2008) was obtained from all participants prior to the recruitment into each wave of the examination. Research within CAIDE has been carried out in compliance with the ethical principles for medical research involving human subjects expressed in the Declaration of Helsinki. This study is reported as per the Strengthening the Reporting of Observational Studies in Epidemiology (STROBE) guidelines (S1 Checklist).

### Data collection and definitions

Data on demographics (e.g., age, sex, and education), lifestyles (e.g., smoking and physical activity), cardiometabolic factors (e.g., body mass index [BMI], plasma glucose, serum total cholesterol, and blood pressure), and medical history (e.g., diabetes and cerebrovascular and

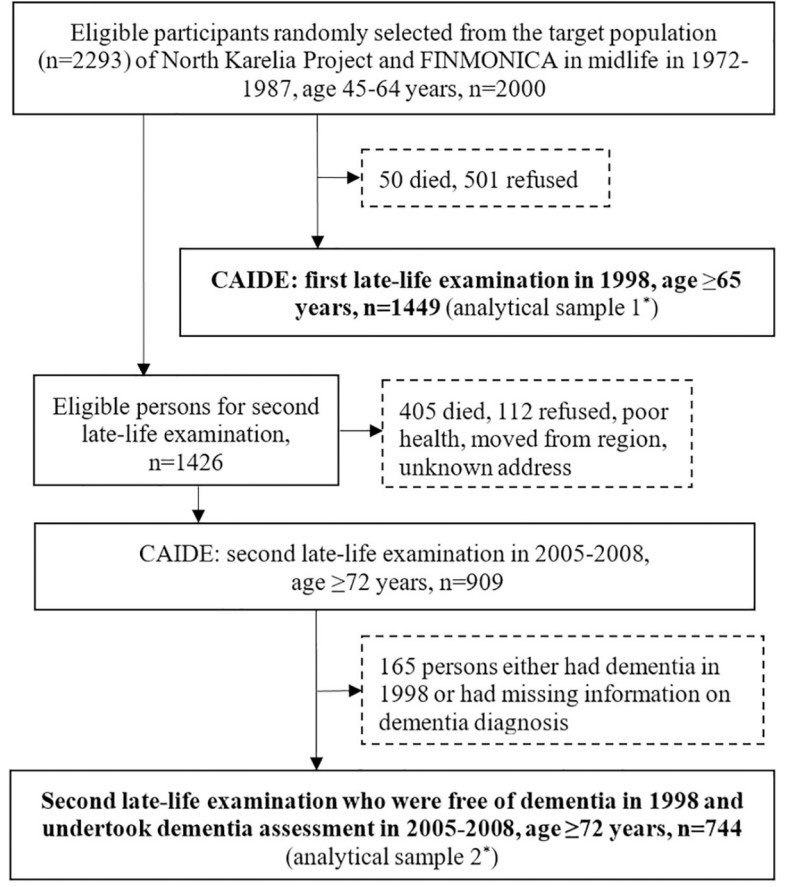

**Fig 1. Flowchart of the study population.** *The analytical sample 1 was used for analyzing the association between midlife CVH metrics and late-life risk of dementia detected both in the 1998 and 2005 to 2008 examinations. The analytical sample 2 was used for analyzing the association between late-life CVH metrics measured in the 1998 examination as well as the patterns of CVH metrics from midlife to late life (1998) with risk of dementia detected in the 2005 to 2008 examination. CAIDE, Cardiovascular Risk Factors, Aging, and Dementia; CVH, cardiovascular health.

cardiovascular events) were collected through self-administered questionnaire survey, clinical examinations, the patient register, the prescribed drug register, and laboratory tests [21,25]. Cardiovascular diseases included angina pectoris, myocardial infarction, heart failure, and stroke diagnosed by physicians [21,25]. Apolipoprotein E (APOE) genotype was analyzed using polymerase chain reaction and *HhaI* restriction enzyme digestion [25] and was dichotomized as carriers versus non-carriers of the ε4 allele.

## Definition of CVH metrics

We defined and scored the composite global CVH metrics based on 6 of the 7 components following the AHA's recommendations [11], with some modifications, owing mainly to the lack of data on diet in both midlife and late life as well as the lack of data on fasting plasma glucose in midlife. The details of CVH metric definitions used in our study and the comparisons with the definitions from AHA were described in Table 1. Briefly, we categorized each of the CVH metric components into poor level (score = 0), intermediate level (score = 1), and ideal level (score = 2). The ideal level of each of the 6 CVH metrics was defined as follows. Ideal smoking status was defined as never smoking or having quitted for more than 1 year. Ideal physical activity was defined as having ≥2 times per week of leisure time physical activity that lasted at

least 20 to 30 minutes and caused breathlessness and sweating [26]. Ideal BMI was defined as a BMI <25 kg/m$^2$. The AHA recommended cutoffs for blood glucose and total cholesterol were converted to the units that we normally used. Ideal blood glucose in late life was defined as fasting plasma glucose <5.6 mmol/l (corresponding to <100 mg/dl in the AHA's recommendations) without treatment. However, due to the lack of data on fasting plasma glucose in midlife, information on self-reported history of diabetes, a recorded diagnosis of diabetes from the patient register, and the use of antidiabetic medication from the prescribed drug register were used as a proxy. Thus, ideal level of blood glucose in midlife was defined as having none of the self-reported history of diabetes, registered diagnosis of diabetes, and the use of prescribed antidiabetic medication. Ideal total cholesterol was defined as untreated serum total cholesterol <5.17 mmol/l (corresponding to <200 mg/dl in the AHA's recommendations). Ideal blood pressure was defined as systolic blood pressure (SBP) <120 mmHg and diastolic blood pressure (DBP) <80 mmHg without antihypertensive treatment.

Then, the global CVH metric score was calculated as the sum of scores of 6 CVH metric components, and the global CVH metrics were categorized into poor (total score ≤5), intermediate (6 to 7), and ideal (≥8) levels. In addition, smoking, physical activity, and BMI were used to define behavioral CVH metrics, and fasting plasma glucose, serum total cholesterol, and blood pressure as biological CVH metrics [11,16]. The levels of composite behavioral and biological CVH metrics were categorized into poor (a composite score ≤2), intermediate (3 to 4), and ideal (≥5) levels, respectively.

## Clinical diagnosis of dementia

Comprehensive cognitive testing was performed at the late-life examinations in 1998 and 2005 to 2008. The Mini-Mental State Examination (MMSE) was used to assess global cognitive

**Table 1. Definitions of CVH metrics in the recommendations of AHA and in the study of CAIDE.**

| Metrics | Poor level (score = 0) | Intermediate level (score = 1) | Optimal level (score = 2) |
|---|---|---|---|
| Smoking* | Current smoking | Former smoking ≤1 year ago | Never smoking or quit >1 year |
| Physical activity | No physical activity | AHA: 1–149 min/week moderate intensity or 1–74 min/week vigorous intensity or 1–149 min/week moderate + vigorous<br>CAIDE: ≤1 time/week of leisure time physical activity that lasts at least 20–30 min and causes breathlessness and sweating | AHA: ≥150 min/week moderate intensity or ≥75 min/week vigorous intensity or ≥150 min/week moderate + vigorous<br>CAIDE: ≥2 times/week of leisure time physical activity that lasts at least 20–30 min and causes breathlessness and sweating |
| Diet | AHA: 0–1 component of a healthy diet score<br>CAIDE: No data | AHA: 2–3 components of a healthy diet score<br>CAIDE: No data | AHA: 4–5 components of a healthy diet score<br>CAIDE: No data |
| BMI* | ≥30 kg/m$^2$ | 25–29.9 kg/m$^2$ | <25 kg/m$^2$ |
| Fasting plasma glucose# | AHA: ≥7.0 mmol/l<br>CAIDE: diabetes treated on drugs (midlife) | AHA: 5.6–6.9 mmol/l or treated to goal<br>CAIDE: diabetes treated on diet (midlife) | AHA: <5.6 mmol/l<br>CAIDE: no diabetes (midlife) |
| Serum total cholesterol* | ≥6.21 mmol/l | 5.17–6.20 mmol/l or treated to goal | <5.17 mmol/l untreated |
| Blood pressure* | SBP ≥140 mmHg or DBP ≥90 mmHg | SBP 120–139 or DBP 80–89 mmHg or treated to goal | SBP <120 mmHg and DBP <80 mmHg untreated |

* The definitions for these components in both midlife and late life used in our study were the same as those from AHA's recommendations (Lloyd-Jones et al., 2010).
# The definition for late-life plasma glucose was the same as that from AHA. In midlife, due to lack of fasting plasma glucose, a proxy was used including the information on self-reported history of diabetes, a recorded diagnosis of diabetes from the inpatient register, and the use of antidiabetic medication from the prescribed drug register.

AHA, American Heart Association; BMI, body mass index; CAIDE, Cardiovascular Risk Factors, Aging, and Dementia; CVH, cardiovascular health; DBP, diastolic blood pressure; SBP, systolic blood pressure.

function. Dementia was ascertained following a 3-step protocol that included a screening phase, a clinical phase, and a differential diagnostic phase [21,22,27]. Briefly, in the 1998 examination, participants with an MMSE score ≤24 were referred to the clinical phase for further evaluations. In the 2005 to 2008 examination, participants with either an MMSE score ≤24, or a decrease in the MMSE score since the 1998 examination ≥3 points, or <70% delayed recall in the Consortium to Establish a Registry for Alzheimer's Disease word list, or informant's serious concerns regarding the participant's cognition, were referred to the clinical phase. In both the 1998 and 2005 to 2008 examinations, the clinical phase included thorough physical examination and neuropsychological testing. If the person was suspected to have dementia, further evaluations were made, including laboratory blood tests, a chest radiograph, an electrocardiogram test, brain MRI/CT scans, and a cerebrospinal fluid analysis as needed. The final diagnoses of dementia were made by a review board after careful evaluation of all available information following the criteria outlined in the Diagnostic and Statistical Manual of Mental Disorders, Fourth Edition (DSM-IV).

## Statistical analysis

The study protocol and analytical plan are reported in Supporting information (S1 Study protocol). The midlife (1972 to 1987) and late-life (1998) characteristics were compared between participants included in the analytical sample and those lost to follow-up by using Student $t$ test for continuous variables and chi-squared test for categorical variables. Incidence of dementia was calculated as the number of dementia cases developed during the follow-up period divided by the total person-years of follow-ups. Cox proportional hazards models were used to estimate the hazard ratio (HR) and 95% confidence interval (CI) of dementia associated with global, behavioral, and biological CVH metrics in midlife and late life as well as the patterns of CVH metrics from midlife to late life, in which the follow-up time was used as the time scale. When examining midlife CVH metrics in relation to risk of dementia detected in 1998 and 2005 to 2008, we estimated the follow-up time from the date of midlife examination to the date of dementia diagnosis or the last contact in the 1998 or 2005 to 2008 examination. When examining the associations of late-life CVH metrics measured in 1998 and CVH metric patterns from midlife to late life (1998) with risk of dementia detected in 2005 to 2008, we estimated the follow-up time from the date of the 1998 examination to the date of the 2005 to 2008 examination. The proportional hazards assumption was verified using time-dependent coefficient in the Cox regression models. To account for possible influence of selective survival, we estimated the cumulative incidence of dementia from Cox proportional hazards models while taking into account death as a competing risk event [28].

We reported the main results from 3 models. Model 1 was adjusted for age, sex, and education, and in model 2, we further adjusted for baseline cardiovascular disease and APOE genotype. We considered APOE genotype as a potential confounder because APOE ε4 allele is a genetic risk factor for dementia and also the APOE gene is involved in the lipid transport and lipoprotein metabolism [29,30]. In addition, to assess the potential effect of selective survival, we reported results from the Fine and Gray competing risk regression models (model 3) (S1 Statistical methods) where death was considered as a competing risk event and all covariates in the model 2 were included [31].

IBM SPSS Statistics 26 for Windows (IBM SPSS, Chicago, Illinois) was used for the descriptive analysis and Cox regression analysis. SAS 9.4 Statements (SAS Institute, 2013, Cary, North Carolina, United States of America) was used for estimating the cumulative incidence of dementia and performing the Fine and Gray competing risk regression analysis.

## Results

### Characteristics of study participants in midlife (1972 to 1987) and late life (1998)

Table 2 shows the midlife characteristics of participants in the analytical sample 1 as well as the comparison of characteristics in both midlife and late life between participants included in the analytical sample 2 ($n = 744$) and those not included ($n = 705$). Compared with those not included in the analytical sample 2, the included participants in midlife were younger ($p < 0.001$), more likely to be female ($p = 0.031$), more educated ($p < 0.001$), had higher proportions of ideal smoking ($p = 0.002$), ideal BMI ($p = 0.004$), and ideal blood pressure ($p < 0.001$), a higher score of global, behavioral, and biological CVH metrics (all $p < 0.001$), and were less likely to carry APOE ε4 allele ($p = 0.003$) and have cardiovascular diseases ($p = 0.024$). There was no significant difference in the distributions of physical activity, diabetes, and total cholesterol in midlife between those included and not included. In addition, compared with those not included in the analytical sample 2, the included participants in late life (1998) had higher proportions of ideal smoking ($p < 0.001$), ideal physical activity ($p < 0.001$), ideal blood glucose ($p < 0.001$), higher global and behavioral CVH scores (both $p < 0.001$), and lower prevalence of cardiovascular disease ($p < 0.001$).

### Midlife CVH metrics and risk of incident dementia

Compared with people having poor global CVH metrics in midlife, those with an ideal level of global CVH metrics in midlife had a 48% lower risk of dementia after controlling for potential confounding factors and a 54% reduced risk of dementia after further taking into account death as a competing risk event (Table 3, models 2 and 3). When the global CVH metric score in midlife was analyzed as a continuous variable, the fully adjusted HR of dementia was 0.86 (95% CI: 0.76, 0.98; $p = 0.019$) for per 1-point increment in the CVH metric score, and the HR remained unchanged in model 3 when death as a competing risk event was taken into account. In addition, we analyzed the association of behavioral and biological CVH metrics in midlife with dementia risk in late life. The associations of continuous behavioral CVH metric score and the ideal level of behavioral CVH metrics with a reduced risk of dementia were statistically evident, especially in model 3 when the competing risk due to death was taken into consideration. In contrast, the tendency of an association between ideal biological CVH metrics in midlife and the reduced dementia risk disappeared in the competing risk models (Table 3, models 1 to 3).

Furthermore, of the 6 individual CVH metric components in midlife, intermediate and ideal smoking (versus poor) and ideal BMI (versus poor) were significantly associated with the reduced risk of dementia (S1 Table, model 2). However, the association was no longer significant after taking into account death as a competing risk event (S1 Table, model 3).

### Late-life CVH metrics and risk of incident dementia

Compared with poor CVH metrics in late life, having an intermediate or ideal level of global, behavioral, and biological CVH metrics was not significantly associated with the HR of dementia in models 1 and 2, but when the competing risk due to death was taken into account, an ideal level of biological CVH metrics was significantly associated with an increased HR of dementia (Table 4). Similarly, as a continuous variable, none of the global, behavioral, and biological CVH metric scores in late life was significantly associated with dementia risk in models 1 and 2, but after taking into account death as a competing risk event, per 1-point increment in late-life biological CVH metric score was significantly associated with an over 40% increased

**Table 2. Characteristics of study participants in midlife (1972 to 1987) and late life (1998).**

| Characteristics* | Analytical sample 1, midlife (n = 1449) | Analytical sample 2 in midlife, 1972–1987 | | | Analytical sample 2 in late life, 1998 | | |
|---|---|---|---|---|---|---|---|
| | | Excluded (n = 705) | Included (n = 744) | p | Excluded (n = 705) | Included (n = 744) | p |
| Age (years), mean (SD) | 50.4 (6.0) | 51.4 (5.9) | 49.5 (5.9) | <0.001 | 72.5 (4.2) | 70.2 (3.5) | <0.001 |
| Women | 900 (62.1) | 418 (59.3) | 482 (64.8) | 0.031 | 418 (59.3) | 482 (64.8) | 0.031 |
| Education (years), mean (SD) | 8.6 (3.4) | 7.9 (3.3) | 9.2 (3.5) | <0.001 | 7.9 (3.3) | 9.2 (3.5) | <0.001 |
| Smoking | | | | | | | |
| Poor | 61 (4.2) | 39 (5.5) | 22 (3.0) | 0.002 | 58 (8.5) | 34 (4.6) | <0.001 |
| Intermediate | 290 (20.0) | 158 (22.4) | 132 (17.7) | | 221 (32.3) | 198 (26.8) | |
| Ideal | 1,098 (75.8) | 508 (72.1) | 590 (79.3) | | 406 (59.3) | 506 (68.6) | |
| Physical activity | | | | | | | |
| Poor | 51 (3.6) | 30 (4.4) | 21 (2.9) | 0.321 | 73 (10.5) | 27 (3.6) | <0.001 |
| Intermediate | 772 (54.4) | 373 (54.1) | 399 (54.6) | | 146 (21.1) | 123 (16.6) | |
| Ideal | 597 (42.0) | 286 (41.5) | 311 (42.5) | | 473 (68.4) | 593 (79.8) | |
| BMI | | | | | | | |
| Poor | 241 (16.6) | 139 (19.7) | 102 (13.7) | 0.004 | 202 (28.7) | 192 (25.8) | 0.133 |
| Intermediate | 718 (49.6) | 347 (49.2) | 371 (49.9) | | 321 (45.5) | 378 (50.8) | |
| Ideal | 490 (33.8) | 219 (31.1) | 271 (36.4) | | 182 (25.8) | 174 (23.4) | |
| Plasma glucose# | | | | | | | |
| Poor | 55 (3.8) | 32 (4.5) | 23 (3.1) | 0.071 | 65 (9.2) | 31 (4.2) | <0.001 |
| Intermediate | 3 (0.2) | 3 (0.4) | 0 (0.0) | | 97 (13.8) | 90 (12.1) | |
| Ideal | 1391 (96.0) | 670 (95.0) | 721 (96.9) | | 543 (77.0) | 623 (83.7) | |
| Total cholesterol | | | | | | | |
| Poor | 944 (65.1) | 476 (67.5) | 468 (62.9) | 0.103 | 229 (32.5) | 265 (35.6) | 0.095 |
| Intermediate | 384 (26.5) | 169 (24.0) | 215 (28.9) | | 343 (48.7) | 369 (49.6) | |
| Ideal | 121 (8.4) | 60 (8.5) | 61 (8.2) | | 133 (18.9) | 110 (14.8) | |
| Blood pressure | | | | | | | |
| Poor | 618 (42.7) | 337 (47.8) | 281 (37.8) | <0.001 | 276 (39.1) | 276 (37.1) | 0.692 |
| Intermediate | 760 (52.4) | 339 (48.1) | 421 (56.6) | | 415 (58.9) | 451(60.6) | |
| Ideal | 71 (4.9) | 29 (4.1) | 42 (5.6) | | 14 (2.0) | 17 (2.3) | |
| Global CVH metrics score, mean (SD) | 7.2 (1.5) | 7.0 (1.5) | 7.4 (1.5) | <0.001 | 7.1 (1.7) | 7.6 (1.5) | <0.001 |
| Behavioral CVH metrics score, mean (SD) | 4.2 (1.0) | 4.1 (1.1) | 4.4 (1.0) | <0.001 | 4.2 (1.1) | 4.4 (1.0) | <0.001 |
| Biological CVH metrics score, mean (SD) | 3.0 (1.0) | 2.9 (1.0) | 3.1 (1.0) | <0.001 | 3.2 (1.1) | 3.2 (1.0) | 0.217 |
| APOE ε4 allele carriers | 499 (35.1) | 265 (39.0) | 234 (31.5) | 0.003 | 265 (39.0) | 234 (31.5) | 0.003 |
| Cardiovascular diseases | 114 (7.9) | 67 (9.5) | 47 (6.3) | 0.024 | 260 (37.6) | 190 (25.5) | <0.001 |

Data were n (%), unless otherwise specified.

* The number of participants with missing values in analytical sample 1 was 21 persons for education, 27 persons for APOE genotype, and 29 persons for midlife physical activity, and in analytical sample 2 was 11 persons for education, 1 person for APOE genotype, 6 persons for late-life smoking, and 1 person for late-life physical activity. In the subsequence analysis, a dummy variable was created to indicate the missing values for education and APOE genotype, whereas missing value on physical activity and smoking was considered as null (i.e., poor level) in the calculation of global CVH metric score.

# In midlife, information on fasting plasma glucose was lacking, instead, the information on self-reported diabetes, a recorded diagnosis of diabetes from the inpatient register, and the use of antidiabetic medication from the prescribed drug register were used as a proxy.

APOE, apolipoprotein E; BMI, body mass index; CVH, cardiovascular health.

risk of dementia (HR = 1.41; 95% CI = 1.02, 1.94; p = 0.036) (Table 4). In addition, none of the 6 individual CVH metric components in late life was significantly associated with the risk of dementia in any of the 3 models (S2 Table).

**Table 3. The association of CVH metrics in midlife (1972 to 1987) and risk of dementia detected in late life (both 1998 and 2005 to 2008) (*n* = 1,449).**

| Midlife CVH metrics (score) | No. of participants | No. of dementia cases | Person-years of follow-up | Incidence (per 1,000 person-years) | 20-year cumulative incidence (%)* | Model 1# | | Model 2# | | Model 3# | |
|---|---|---|---|---|---|---|---|---|---|---|---|
| | | | | | | HR (95% CI) | *p* | HR (95% CI) | *p* | HR (95% CI) | *p* |
| **Global CVH metrics** | | | | | | | | | | | |
| Per 1-point increment | 1,449 | 108 | 30,680 | 3.52 | 6.16 | 0.86 (0.76, 0.97) | 0.016 | 0.86 (0.76, 0.98) | 0.019 | 0.86 (0.76, 0.97) | 0.014 |
| Poor (≤5) | 183 | 28 | 3,619 | 7.74 | 9.03 | 1.00 (reference) | | 1.00 (reference) | | 1.00 (reference) | |
| Intermediate (6–7) | 645 | 52 | 13,904 | 3.74 | 6.35 | 0.69 (0.42, 1.13) | 0.143 | 0.71 (0.43, 1.16) | 0.174 | 0.63 (0.38, 1.04) | 0.072 |
| Ideal (≥8) | 621 | 28 | 13,156 | 2.13 | 3.82 | 0.51 (0.29, 0.90) | 0.021 | 0.52 (0.29, 0.93) | 0.027 | 0.46 (0.27, 0.77) | 0.003 |
| **Behavioral CVH metrics** | | | | | | | | | | | |
| Per 1-point increment | 1,449 | 108 | 30,680 | 3.52 | 6.16 | 0.86 (0.72, 1.02) | 0.083 | 0.86 (0.73, 1.02) | 0.090 | 0.81 (0.69, 0.96) | 0.013 |
| Poor (≤2) | 85 | 13 | 1,612 | 8.06 | 5.71 | 1.00 (reference) | | 1.00 (reference) | | 1.00 (reference) | |
| Intermediate (3–4) | 741 | 65 | 15,865 | 4.10 | 7.24 | 1.47 (0.73, 2.94) | 0.278 | 1.45 (0.72, 2.91) | 0.294 | 0.67 (0.36, 1.23) | 0.195 |
| Ideal (≥5) | 623 | 30 | 13,204 | 2.27 | 3.54 | 0.73 (0.35, 1.53) | 0.409 | 0.73 (0.35, 1.52) | 0.402 | 0.42 (0.22, 0.83) | 0.012 |
| **Biological CVH metrics** | | | | | | | | | | | |
| Per 1-point increment | 1,449 | 108 | 30,680 | 3.52 | 6.16 | 0.81 (0.65, 1.00) | 0.052 | 0.82 (0.66, 1.01) | 0.063 | 0.90 (0.71, 1.14) | 0.374 |
| Poor (≤2) | 464 | 48 | 9,760 | 4.92 | 7.90 | 1.00 (reference) | | 1.00 (reference) | | 1.00 (reference) | |
| Intermediate (3–4) | 878 | 55 | 18,768 | 2.93 | 4.73 | 0.73 (0.49, 1.09) | 0.124 | 0.74 (0.49, 1.10) | 0.140 | 0.83 (0.56, 1.24) | 0.363 |
| Ideal (≥5) | 107 | 5 | 2,152 | 0.93 | 2.97 | 0.62 (0.24, 1.64) | 0.338 | 0.63 (0.24, 1.67) | 0.353 | 1.04 (0.41, 2.62) | 0.941 |

* The cumulative incidence was calculated from the crude Cox models after taking into account the competing risk of death.

# Model 1 was adjusted for age, sex, and education; model 2 was additionally adjusted for APOE ε4 allele and cardiovascular disease in midlife; and model 3 included death as a competing risk event with the adjustment of all covariates in model 2.

APOE, apolipoprotein E; CI, confidence interval; CVH, cardiovascular health; HR, hazard ratio.

## Patterns of CVH metrics from midlife to late life and risk of incident dementia

When midlife and late-life global CVH metric scores were entered simultaneously into the model, the fully adjusted HR of dementia associated with per 1-point increment in CVH score in midlife and late life was 0.82 (95% CI: 0.66, 1.03; *p* = 0.082) and 1.21 (0.97, 1.51; *p* = 0.093), respectively. After further taking into account death as a competing risk event, the HR remained unchanged for both midlife and late-life global CVH metric scores. There was a statistical interaction between midlife global CVH metric score and late-life global CVH metric score on the risk of dementia (*p* for interaction = 0.001).

Fig 2 showed the HRs (95% CI) of dementia associated with the combinations (or patterns) of different levels of global, behavioral, and biological CVH metrics in midlife and late life, controlling for multiple potential confounding factors. (1) Global CVH metrics: Compared with persons having a poor level of global CVH metrics in both midlife and late life, those with

**Table 4. The association of CVH metrics in late life (1998) and risk of dementia detected in late life (2005 to 2008) (*n* = 744).**

| Late-life CVH metrics (score) | No. of participants | No. of dementia cases | Person-years of follow-up | Incidence (per 1,000 person-years) | 8-year cumulative incidence (%)* | Model 1# | | Model 2# | | Model 3# | |
|---|---|---|---|---|---|---|---|---|---|---|---|
| | | | | | | HR (95% CI) | *p* | HR (95% CI) | *p* | HR (95% CI) | *p* |
| **Global CVH metrics** | | | | | | | | | | | |
| Per 1-point increment | 744 | 47 | 6,168 | 7.62 | 4.24 | 1.04 (0.86, 1.26) | 0.700 | 1.09 (0.89, 1.33) | 0.409 | 1.10 (0.87, 1.39) | 0.433 |
| Poor (≤5) | 66 | 6 | 537 | 11.17 | 6.64 | 1.00 (reference) | | 1.00 (reference) | | 1.00 (reference) | |
| Intermediate (6–7) | 276 | 14 | 2,259 | 6.20 | 3.63 | 0.52 (0.20, 1.38) | 0.191 | 0.60 (0.22, 1.69) | 0.338 | 0.61 (0.23, 1.64) | 0.329 |
| Ideal (≥8) | 402 | 27 | 3,372 | 8.01 | 4.33 | 0.70 (0.29, 1.73) | 0.444 | 0.91 (0.34, 2.41) | 0.850 | 0.96 (0.35, 2.58) | 0.929 |
| **Behavioral CVH metrics** | | | | | | | | | | | |
| Per 1-point increment | 744 | 47 | 6,168 | 7.62 | 4.24 | 0.89 (0.69, 1.15) | 0.367 | 0.93 (0.71, 1.22) | 0.608 | 0.86 (0.62, 1.19) | 0.368 |
| Poor (≤2) | 36 | 5 | 299 | 16.72 | 8.88 | 1.00 (reference) | | 1.00 (reference) | | 1.00 (reference) | |
| Intermediate (3–4) | 353 | 19 | 2,909 | 6.53 | 3.90 | 0.53 (0.20, 1.45) | 0.218 | 0.53 (0.19, 1.44) | 0.212 | 0.50 (0.15, 1.66) | 0.258 |
| Ideal (≥5) | 355 | 23 | 2,960 | 7.77 | 4.21 | 0.63 (0.23, 1.72) | 0.367 | 0.70 (0.25, 1.93) | 0.491 | 0.57 (0.17, 1.95) | 0.371 |
| **Biological CVH metrics** | | | | | | | | | | | |
| Per 1-point increment | 744 | 47 | 6,168 | 7.62 | 4.24 | 1.24 (0.93, 1.64) | 0.140 | 1.28 (0.96, 1.70) | 0.092 | 1.41 (1.02, 1.94) | 0.036 |
| Poor (≤2) | 171 | 7 | 1,394 | 5.02 | 2.68 | 1.00 (reference) | | 1.00 (reference) | | 1.00 (reference) | |
| Intermediate (3–4) | 495 | 31 | 4,102 | 9.02 | 4.38 | 1.71 (0.73, 4.00) | 0.215 | 2.06 (0.83, 5.14) | 0.120 | 2.10 (0.84, 5.20) | 0.110 |
| Ideal (≥5) | 78 | 9 | 673 | 13.37 | 6.63 | 2.38 (0.86, 6.64) | 0.097 | 2.85 (0.96, 8.46) | 0.059 | 3.54 (1.16, 10.83) | 0.027 |

* The cumulative incidence was calculated from the crude Cox models after taking into account the competing risk of death.

# Model 1 was adjusted for age, sex, and education; model 2 was additionally adjusted for APOE ε4 allele and cardiovascular disease in late life; and model 3 included death as a competing risk event with the adjustment of all covariates in model 2.

APOE, apolipoprotein E; CI, confidence interval; CVH, cardiovascular health; HR, hazard ratio.

an intermediate level of global CVH metrics in both midlife and late life and those with midlife ideal and late-life intermediate global CVH metrics had the fully adjusted HR of 0.25 (95% CI: 0.08, 0.86; *p* = 0.028) and 0.14 (0.02, 0.76; *p* = 0.024), respectively, for dementia (Fig 2A). However, the HRs were not significant when taking into account death as a competing risk event. There was no significant association with dementia risk for any other combinations of midlife and late-life CVH metrics. (2) Behavioral CVH metrics: Compared with people with the poor level of behavioral CVH metrics in both midlife and late life, those with an intermediate level of behavioral CVH metrics in both midlife and late life, an intermediate level in midlife but an ideal level in late life, an ideal level in midlife but an intermediate level in late life, and an ideal level in both midlife and late life had a significantly reduced risk of dementia (Fig 2B). The associations remained significant after taking into account death as a competing risk event. We found no significant association between other combining groups of behavioral CVH metrics and risk of dementia. (3) Biological CVH metrics: Of all the 9 combinations of biological

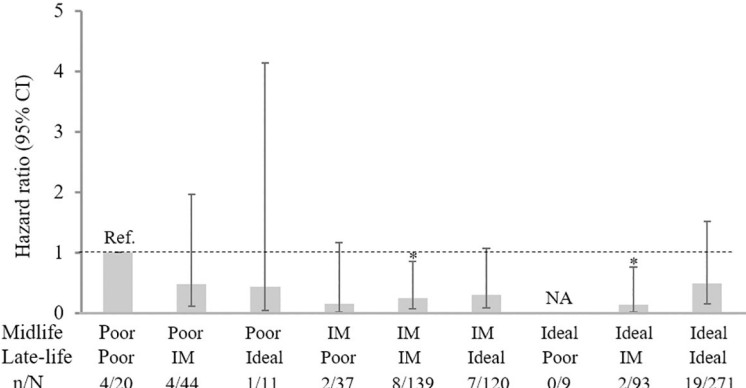

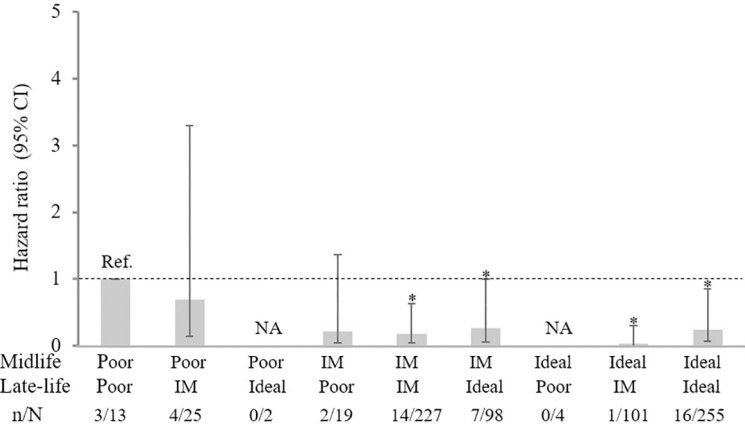

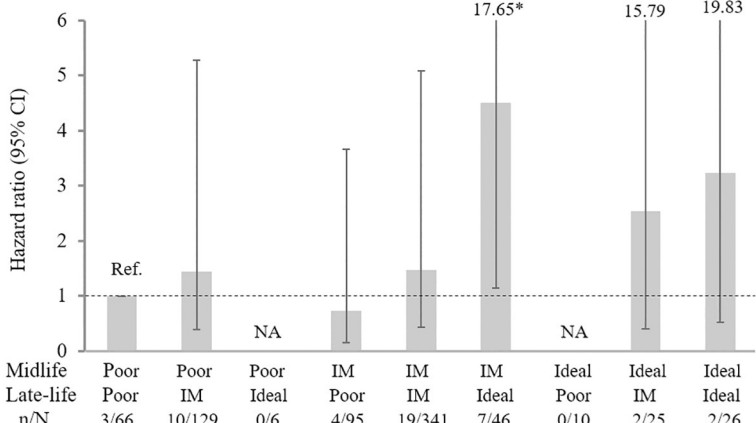

**Fig 2. The HR (95% CI) of dementia associated with the joint patterns of midlife and late-life CVH metrics: A, global CVH metrics; B, behavioral CVH metrics; C, biological CVH metrics.** HR (95% CI) was derived from the Cox proportional hazards models controlling for age, sex, education, APOE ε4 allele, and cardiovascular disease in midlife. *$p < 0.05$. APOE, apolipoprotein E; CVH, cardiovascular health; CI, confidence interval; HR, hazard ratio; IM, intermediate; NA, not available; n/N, the number of dementia cases/number of participants.

CVH metric levels in midlife and late life, only the combination of midlife intermediate and late-life ideal levels of biological CVH metrics was significantly associated with an increased risk of dementia in the fully adjusted Cox regression models (Fig 2C). However, when taking into account death as a competing risk event, compared with people having the poor level of biological CVH metrics in both midlife and late life, the HR of dementia was 7.62 (95% CI: 1.97, 29.48; *p* = 0.003) for those with midlife intermediate level and late-life ideal level of biological CVH metrics and 7.30 (1.22, 43.78; *p* = 0.030) for those with midlife ideal level and late-life intermediate level of biological CVH metrics.

## Discussion

In this population-based cohort study, we found that compared with poor CVH metrics in midlife, an ideal level of midlife global CVH metrics, and behavioral CVH metrics in particular, was associated with a lower risk of dementia in late life. There was no association between CVH metrics and dementia risk when CVH metrics were assessed in late life. However, when the global CVH metric patterns from midlife to late life were examined, compared with the poor global CVH metrics in both midlife and late life, having an intermediate or ideal level of global CVH metrics in midlife or late life was associated with a reduced risk of dementia, although the competing risk due to death could partly account for the associations (Fig 2A). Notably, the association of an intermediate or ideal level of behavioral CVH metrics in either midlife or late life with a reduced risk of dementia was present independent of competing risk and all potential confounders examined (Fig 2B). Finally, having an ideal or intermediate level of late-life biological CVH metrics in combination with either an ideal or intermediate biological CVH metrics in midlife (versus poor level both in midlife and late life) was associated with an increased risk of dementia (Fig 2C).

Population-based studies have rarely investigated the association of composite CVH metrics with the risk of dementia from a life-course perspective. The Whitehall II cohort study, which used the modified AHA's definitions to define smoking and diet, showed that per 1-point increment in the global CVH metric score at the age of 50 years was associated with an over 10% decreased risk of dementia during an over 25-year follow-up (HR = 0.89; 95% CI = 0.85 to 0.95) [16]. Similarly, the Atherosclerosis Risk in Communities (ARIC) cohort study suggested that a higher global CVH metric score in middle age (mean age, 54 years) was associated with slower cognitive decline over a 20-year follow-up period [32]. In the Finnish CAIDE cohort, we found that having a higher global CVH metric score or having an ideal level of global CVH metrics in midlife (mean age, 50.4 years) was associated with the lower risk of dementia. Taken together, these studies provide consistent evidence supporting the view that achieving optimal CVH in midlife is crucial for reducing late-life dementia risk.

We found no evidence for the association between late-life global CVH metrics and the risk of dementia, which was consistent with the findings from the population-based study of adults (age ≥55 years) from the health insurance settings in Germany [19]. However, the population-based Three-City Study of older adults (age ≥65 years) in France did show that having more optimal CVH metrics or having a higher CVH metric score defined following the AHA's recommendations was associated with a lower risk of dementia [17]. The inconsistent findings across studies might be partially attributable to differences in methodology (e.g., lack of diet data in our study) and characteristics of study population (e.g., age).

This discrepancies between associations of midlife and late-life CVH metrics with the risk of dementia were in accordance with the fact that the relationships of cardiometabolic components of CVH metrics (e.g., BMI in behavioral CVH metrics and blood pressure and total cholesterol in biological CVH metrics) with the risk of dementia vary with age; the strength of

associations between these cardiometabolic risk factors and risk of dementia is attenuated with age from young adulthood and midlife onwards, and may even be reversed among very old people [33–35]. Thus, these individual CVH metrics may render the global CVH metrics less predictive for dementia risk as people age [19]. Indeed, we found that an ideal level of late-life biological CVH metrics was associated with an elevated risk of dementia. However, this association may partly reflect the reverse causality because the ideal (or low) levels of certain biological CVH metric components in late life (e.g., blood pressure <120/80 mmHg and low total cholesterol) might be a marker of preclinical dementia [8,10,33]. This should be kept in mind when interpreting the results from studies that involve late-life composite global and biological CVH metrics in relation to dementia risk.

The association of CVH metric patterns from midlife to late life with the risk of dementia has been rarely evaluated so far in the population-based settings. We found that compared with poor levels of global CVH metrics in both midlife and late life, having the intermediate or ideal level of global CVH metrics, especially behavioral CVH metrics, in either midlife or late life was associated with a substantially reduced risk of dementia. Given the potentially reverse causality of late-life ideal levels of biological components in the global CVH metrics and dementia risk, interpretation of the results involving late-life global CVH metrics needs to be cautious. Our findings implied that maintaining a life-long CVH, and behavioral health in particular, from midlife onwards may help achieve healthy brain aging. This supports the view that intervention strategies to promote CVH, especially behavioral health, from the life-course prospective might help prevent or delay the onset of dementia [2]. Because the CVH metrics are modifiable and manageable, our findings have significant implications for public health. This is particularly relevant given that dementia has become a global public health priority in our aging society.

The mechanisms underlying the association between the CVH metrics over the life course and risk of dementia are not fully understood. The poor levels (as risk factors) of CVH metrics, especially occurring from midlife, are supposed to contribute mainly to macro- and microvascular lesions and neurodegenerative process in the brain, e.g., oxidative stress, inflammation, atherosclerosis, hypoxia, cerebral small vessel lesions, and advanced glycation end products [36]. Optimal or ideal levels of CVH metrics (e.g., no smoking, physical activity, and normal blood glucose) are associated with less burden of cerebrovascular damage [37], which then leads to fewer white matter lesions and brain infarcts as well as less severe neurodegeneration [2,38]. The optimal brain health resulting from optimal CVH profiles may in turn contribute to the lower risk of cognitive impairment [39] and cognitive decline [17,18], and thus lead to a lower risk of dementia.

Strengths of this study include the design of population-based cohort study with long follow-up time from midlife to late life as well as the comprehensive assessments for the diagnosis of dementia. However, our study also has limitations. First, we used the modified AHA's definitions to assess CVH metrics owing to the lack of data on diet in both midlife and late life as well as midlife plasma glucose. Lack of dietary data might underestimate the potential protective effect of ideal CVH metrics against dementia because data from a small subsample of the CAIDE participants (n = approximately 350) showed that midlife healthy diet and beneficial dietary changes from midlife to late life were associated with a reduced risk of dementia [40,41]. Similarly, the lack of data on midlife fasting plasma glucose might have misclassified some people with abnormal plasma glucose into the ideal category, and thus underestimate the possible beneficial effect of the ideal midlife CVH metrics on dementia risk. Second, the analytic sample was generally healthier compared to dropouts (e.g., death, refusal, or missing information for the diagnosis of dementia). Thus, the selective survival by both CVH metric levels and cognitive conditions (e.g., cognitive impairment and dementia) over the follow-up

period from midlife to late life might have led to an underestimation of the true associations between poor CVH metrics and risk of dementia because people with poor CVH metrics or poor cognitive conditions were less likely to survive into old age [21]. However, we have estimated the cumulative incidence of dementia and also used the competing risk models to partially account for the impact of selective survival on the associations between CVH metrics and risk of dementia. Nevertheless, cautiousness is needed when generalizing our study findings to the entire middle-aged and elderly population. Finally, the statistical power was limited for the analysis of some subgroups due to small numbers of participants and dementia cases, especially in the analysis of midlife and late-life CVH metric patterns and risk of dementia. Thus, our findings deserve further confirmation in the large-scale population-based studies.

In conclusion, in this population-based cohort study, we observed that having an ideal or intermediate level of global CVH metrics, especially the behavioral CVH metrics, from midlife to later in life is associated with a lower risk of dementia. Findings from this study reinforce the view that maintaining a life-long optimal CVH profile, and behavioral health profile in particular, may help reduce the late-life risk of dementia.

## Supporting information

**S1 Checklist. STROBE checklist.** STROBE, Strengthening the Reporting of Observational Studies in Epidemiology.
(DOCX)

**S1 Study protocol.**
(DOCX)

**S1 Statistical methods. Fine and Gray competing risk regression model.**
(DOCX)

**S1 Table. The association of individual CVH metrics in midlife (1972–1987) with risk of dementia detected in late life (both 1998 and 2005–2008) ($n$ = 1,449).** *The number of participants with missing data was 13 persons for physical activity, and these individuals were included in the analysis by creating a dummy variable to indicate those with missing values. #In midlife, information on fasting plasma glucose was lacking, instead, the information on self-reported history of diabetes, a recorded diagnosis of diabetes from the inpatient register, and the use of antidiabetic medication from the prescribed drug register were used as a proxy. §Model 1 was adjusted for age, sex, education, and other components in the table; model 2 was additionally adjusted for APOE ε4 allele and cardiovascular disease in midlife; and model 3 included death as a competing risk event with the adjustment of all covariates in model 2. APOE, apolipoprotein E; CI, confidence interval; CVH, cardiovascular health; HR, hazard ratio.
(DOCX)

**S2 Table. The association of individual CVH metrics in late life (1998) with risk of dementia detected in late life (2005–2008) ($n$ = 744).** *The number of participants with missing data was 6 persons for smoking and 1 person for physical activity, and these individuals were included in the analysis by creating a dummy variable for each factor to indicate those with missing values. #Model 1 was adjusted for age, sex, education, and other components in the table; model 2 was additionally adjusted for APOE ε4 allele and cardiovascular disease in late life; and model 3 included death as a competing risk event with the adjustment of all covariates in model 2. APOE, apolipoprotein E; CI, confidence interval; CVH, cardiovascular health; HR,

hazard ratio.
(DOCX)

## Acknowledgments

The authors would like to thank all the CAIDE participants for their time and valuable contribution to the data of this study.

## Author Contributions

**Conceptualization:** Yajun Liang, Tiia Ngandu, Tiina Laatikainen, Hilkka Soininen, Jaakko Tuomilehto, Miia Kivipelto, Chengxuan Qiu.

**Data curation:** Yajun Liang, Tiia Ngandu.

**Formal analysis:** Yajun Liang.

**Funding acquisition:** Tiia Ngandu, Hilkka Soininen, Miia Kivipelto, Chengxuan Qiu.

**Investigation:** Tiia Ngandu, Tiina Laatikainen, Hilkka Soininen, Jaakko Tuomilehto, Miia Kivipelto.

**Methodology:** Yajun Liang, Tiia Ngandu, Tiina Laatikainen, Hilkka Soininen, Jaakko Tuomilehto, Chengxuan Qiu.

**Project administration:** Tiia Ngandu, Miia Kivipelto.

**Software:** Yajun Liang.

**Supervision:** Miia Kivipelto, Chengxuan Qiu.

**Validation:** Yajun Liang.

**Writing – original draft:** Yajun Liang, Chengxuan Qiu.

**Writing – review & editing:** Yajun Liang, Tiia Ngandu, Tiina Laatikainen, Hilkka Soininen, Jaakko Tuomilehto, Miia Kivipelto, Chengxuan Qiu.

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
