## [Editor Report · Decision Letter 0]

21 Apr 2020

Dear Dr Liang, 

Thank you for submitting your manuscript entitled "Cardiovascular health metrics from mid- to late-life and risk of dementia: a population-based study" for consideration by PLOS Medicine.

Your manuscript has now been evaluated by the PLOS Medicine editorial staff [as well as by an academic editor with relevant expertise] and I am writing to let you know that we would like to send your submission out for external peer review.

Kind regards,

Thomas J McBride, PhD,

PLOS Medicine

---

## [Decision Letter · Decision Letter 1]

7 Jul 2020

Dear Dr. Liang,

Thank you very much for submitting your manuscript "Cardiovascular health metrics from mid- to late-life and risk of dementia: a population-based study" (PMEDICINE-D-20-01344R1) for consideration at PLOS Medicine. 

Your paper was evaluated by a senior editor and discussed among all the editors here. It was also discussed with an academic editor with relevant expertise, and sent to four independent reviewers, including a statistical reviewer. The reviews are appended at the bottom of this email and any accompanying reviewer attachments can be seen via the link below:

[LINK]

In light of these reviews, I am afraid that we will not be able to accept the manuscript for publication in the journal in its current form, but we would like to consider a revised version that addresses the reviewers' and editors' comments. Obviously we cannot make any decision about publication until we have seen the revised manuscript and your response, and we plan to seek re-review by one or more of the reviewers. 

We expect to receive your revised manuscript by Jul 28 2020 11:59PM. Please email us (plosmedicine@plos.org) if you have any questions or concerns.

We look forward to receiving your revised manuscript. 

Sincerely,

Emma Veitch, PhD

PLOS Medicine

On behalf of Clare Stone, PhD, Acting Chief Editor,

PLOS Medicine

plosmedicine.org

*In the last sentence of the Abstract Methods and Findings section, please include a brief note about any key limitation(s) of the study's methodology.

*At this stage, we ask that you include a short, non-technical Author Summary of your research to make findings accessible to a wide audience that includes both scientists and non-scientists (I noted this had been provided as a supplementary file, please include the text immediately following the Abstract in your revised manuscript). This text is subject to editorial change and should be distinct from the scientific abstract. Please see our author guidelines for more information: https://journals.plos.org/plosmedicine/s/revising-your-manuscript#loc-author-summary

*We'd suggest using the STROBE guideline to assist/enhance study reporting - if doing this please include the completed STROBE checklist as Supporting Information. Please add the following statement, or similar, to the Methods: "This study is reported as per the Strengthening the Reporting of Observational Studies in Epidemiology (STROBE) guideline (SChecklist)." The STROBE guideline can be found here: http://www.equator-network.org/reporting-guidelines/strobe/. When completing the checklist, please use section and paragraph numbers, rather than page numbers.

*In the Methods section of your paper, please clarify if the study had a prospective protocol or analysis plan? Please state this (either way) early in the Methods section.

Comments from the reviewers:

Reviewer #1: "Cardiovascular health metrics from mid- to late-life and risk of dementia: a population-based study" attempts to quantify the association of cardiovascular health metrics (CVH) with the risk of incident dementia, on 1449 participants from the Finnish Cardiovascular Risk Factors, Aging and Dementia (CAIDE) study. Six AHA metrics (excluding diet) were used to represent CVH, with the DSM-IV criteria used to diagnose dementia, with Cox regression models used for the analysis. It was concluded that either maintaning lifelong good CVH, or improving CVH from mid to late lift, is associated with substantially reduced risk of dementia.

The usage of AHA-based CVH metrics follows prior studies ([17]-[19]) with similar structures, while the DSM-IV criteria for diagnosing dementia is also fairly long-established. The study sample size of 1449 participants is comparable to the three works referenced above (1033-6626 participants), as is the regression analysis. A major contribution of this paper over prior works pertains to the examination of mid-to-late-life CVH metrics, over late-life CVH metrics alone, which has yielded mixed results in the previous works.

This manuscript thus represents a new datapoint in CVH to dementia association, that clearly builds upon a coherent body of previous research. However, while the total number of participants is in the ballpark of previous studies, the number of positive dementia diagnoses is relatively low (61 in 1998, 47 in 2005-08). This is further reflected in the especially low number of valid cases for some stratifications (particularly poor/poor & the novel midlife/latelife analysis, Table 4), and correspondingly wide hazard ratio confidence intervals. 

There also remain some further concerns:

1. The definition of the two late-life examinations (1998, 2005-08) was not very easy to follow in the Study Design section of the main text, possibly due to the explanation being relegated to the secondary caption of the Figure 1 flowchart. This might be shifted to the main text.

2. The likely relatively large number of participants who died before they could be included in the analytical samples, would appear to have possibly affected the statistics/conclusions. From the main text, it would appear that patients who manifested dementia, but died before a follow-up examination, would be ignored. While dementia incidence might be unknown for these cases, it might be appropriate to consider a survival model for dementia/death, or possibly include fraction/age of death at follow-up in the demographics (e.g. as done in "Midlife cardiovascular fitness and dementia: A 44-year longitudinal population study in women", Horder et al., Neurology, 2018)

3. The adjustment for the APOE genotype in the context of CVH/dementia might be briefly motivated in terms of possible protective effect.

4. The definition for "history of cardiovascular disease", as displayed in Table 1, might be specified in greater detail.

5. The full implementation of intervention assumption for population attributable risk (PAR %) in Page 8 might be more fully elaborated. In particularly, what exactly are these interventions, and what is their expected success rate?

6. For Table 4, "Intermediate or ideal (≥6) in either midlife or late-life" might be clarified as "either midlife or late-life only". Moreover, since improved CVH (i.e. poor in mid-life, intermediate or ideal in late-life) is discussed in the main text, it might also be shown in Table 4 (possibly along with intermediate/ideal in mid-life, poor in late-life)

7. While it is stated that "There was no association between CVH metrics and dementia risk when CVH metrics were assessed in late-life" (Page 10), Table 3 would seem to suggest an overall risk reduction, were the Intermediate & Ideal categories combined and compared against Poor; the authors might also comment on Intermediate & Ideal being analyzed separately in Tables 2 & 3, but combined for the analysis in Table 4.

Reviewer #2: In general this is a well-written manuscript. Weaknesses lie in the fact that not all components of the AHA simple 7 score were available, but that a score was still computed and I don't believe that the score in this format has been validated (if it has been, there should be discussion of this). Because of this, I find less value in use of the AHA Simple 7 metric in this paper and think that it would be more valuable to only report the results of individual components.

* In general the writing is clear, but there are a few sentences that are not grammatically correct and would benefit from some editing. In general, writing is good.

* More detail is needed regarding the definition for physical activity more than twice a week. What was defined as physical activity twice a week? Any physical activity (including walking around the block)? Exercise at a gym? Etc. AHA Simple 7 defines this as >=150 min/week of moderate activity or >=75 min/week of vigorous activity or >=150 min/week of moderate and vigorous activity.

* Since the author's score is missing diet and has a proxy measure for fasting plasma glucose, it isn't really the AHA Simple 7 score anymore. Because of this, it may be more prudent to only evaluate individual components of this score and to not include the computed global CVH metrics score, unless this has been validated previously. 

* In Table 1, it looks like participants in analytical sample 2 (late-life) are survivors (slightly more women, much more ideal physical activity, much more ideal total cholesterol), etc. Can authors discuss this further in their manuscript and what this means for their conclusions? There is some discussion of this, but I think that it could be discussed more extensively. 

Reviewer #3: In this population based Finnish study of initially n=1400 individuals in their mid-life, authors investigate the association between (a) mid-life and late life 6-item cardiovascular health (CVH) and incident dementia, and (b) change in 6-item CVH between mid-life and late-life with incident dementia. In that study, mid-life but not late life CVH was related to lower risk of dementia. In addition, patterns of change in CVH including improvement in CVH were related to lower risk of dementia.

Primordial prevention is a highly relevant and emergent topic and there is prior evidence that higher CVH is related to lower risk of dementia. Authors add novelty in addressing whether change in CVH is related to dementia risk.

However, the current analysis suffers from major limitations:

1. The study is of relatively small size (at best n=2000 in mid-life) limiting the power of the analysis, and precision of the estimates as shown by the width of the confidence intervals. In particular, this precludes to study with more granularity the patterns of change in CVH between mid-life and late-life: potentially 9 groups of change exist and here only three were investigated. As a corollary, lost to follow-up is substantial: 25% between mid-life and late-life exam1, and an additional lost to follow-up of half the population between late life exam 1 and late life exam 2, so that the analytic sample is likely to be highly selected. 

2. As acknowledged by the authors, diet was missing in their CVH evaluation which limits comparison with other studies based on the full (e.g. 7 items) CVH scale. The authors should discuss to which extent this missing metric impacts their results.

3. In that study, there was no significant association between late-life CVH and dementia. The authors concentrate the discussion of this finding by a comparison with a German study that did not find any significant association (ref #19). However, their result is inconsistent with a paper cited in ref #17 that reports significant association with dementia, but the authors did not discuss these inconsistencies.

Reviewer #4: The authors assessed mid-life (ages 45-64) cardiovascular health (CVH) in 2,293 Finnish people in 1972-1987, then ascertained incident dementia among 1,449 survivors in 1998. The authors also assessed late-life (ages 65+) CVH in 1,426 of the survivors in 1998, and again ascertained incident dementia among 744 survivors in 2005-2008 who had been free of dementia in 1998. This study design enabled the authors to relate both mid-life and late-life CVH to incident dementia, which is a strength of the study. The authors correctly stated that few studies have assessed late-life CVH, or a life-course perspective of changes in CVH from mid-life to late-life, in relation to incident dementia, and thus their study would represent an important enhancement to the existing literature.

In my opinion, this study is very good. My comments below are not criticisms of the quality of the study, but rather suggestions for how this already good work might be improved.

MAJOR COMMENTS:

1. Methods/Results: In order to further explore components of mid-life and late-life CVH in relation to incident dementia, I suggest the authors consider adding to their investigation models in which all 6 components are added simultaneously as independent variables, rather than the summary CVH score. These models would show the magnitude and direction of the hazard ratio for each component adjusted for the others, and which of the 6 components matter more than others. This could be especially informative for comparing the results pertaining to mid-life CVH (Table 2) and late-life CVH (Table 3), given that body mass index, blood pressure, and cholesterol may behave differently in mid-life versus late-life in relation to dementia risk.

2. Results, Table 1: I suggest the authors add two columns to Table 1 to enhance readers' understanding of the participants' characteristics. The first new column would be the mid-life characteristics of the subset of analytical sample #1 who were not included in analytical sample #2 (705 people). The second new column would be the mid-life characteristics of analytical sample #2 (744 people). These two columns could be placed in between the existing columns for analytical sample #1 and analytical sample #2. These columns would help us see whether the two halves of analytical sample #1 were different, and would also help us see more directly the change in CVH over time from mid-life to late-life in analytical sample #2. Also, in conjunction with these additions to Table 1, the second paragraph of the Results section could be deleted entirely, because the information in that paragraph partially matches what I am suggesting to add to Table 1.

3. Results, Table 4: I suggest the authors consider adding a model that includes the mid-life CVH score (per 1-point increment) and the late-life CVH score (per 1-point increment) simultaneously as independent variables, and also possibly the interaction of those two CVH scores. This model would complement the model based on combinations of mid-life and late-life CVH categories.

4. Discussion: As the authors note in the paragraph about limitations, this study may suffer from selective survival by levels of CVH. I suggest that the authors also recognize and point out that the study may suffer from selective survival by cognitive decline and dementia. One of the main reasons eligible participants may have not participated in dementia assessment is because they were too cognitively impaired to participate at all, or because they had died as a consequence of dementia, which would be unknown to the authors. I agree with the authors that the magnitude of associations of mid-life CVH and dementia may be under-estimated, and I suggest the authors clarify in their statement that this is likely because of a joint effect of selective survival by CVH level (which can be measured) and selective survival by cognitive decline & dementia (which is unknown for non-survivors and cannot be measured). The first problem, selective survival by CVH level, can be assessed directly in the authors' data, and I suggest the authors conduct this assessment and include the findings in the Results section. For analytical sample #1, compare the mid-life characteristics for those participants who were included in the 1998 dementia assessment versus those who were not. And for analytical sample #2, compare the mid-life and late-life (1998) characteristics for those participants who were included in the 2005-2008 dementia assessment versus those who were not. These analyses will enhance the paper and allow the authors to comment more concretely on the possibility that selective survival influenced the hazard ratios. Fortunately, any bias in the hazard ratios due to selective survival is very likely to be conservative, resulting in under-estimated magnitude of association, rather than anti-conservative.

MINOR COMMENTS:

5. Abstract: The opening statement that associations of CVH with dementia "remain to be explored in cohort studies" seems misleading, given that the authors cited a number of relevant prior publications on CVH and dementia or cognitive impairment or cognitive decline from cohort studies (refs 15-19 cited in the Introduction and refs 28 and 36). I suggest the authors come up with a more appropriate statement for the first sentence of the Abstract.

6. Abstract: The authors stated that the cohort study included 1,449 participants followed from mid-life to late-life. I suggest the authors clarify that 1,449 participants were followed from midlife to late-life (1998), and then a subset of 744 were followed further into late-life (2005-2008). Otherwise, readers of the abstract may wonder why there were relatively few cases of incident dementia (47 cases) identified at the 2005-2008 exam, after 61 incident cases were identified at the 1998 exam.

7. Abstract: I suggest the authors consider including hazard ratios of dementia for intermediate CVH in the abstract, in addition to the hazard ratios for ideal CVH.

8. Abstract: The Conclusions sentence seems slightly mis-matched to the findings. I think the findings supported lower risk of dementia among people who had good CVH in mid-life (Table 2), those who had good CVH in either mid-life or late-life (Table 4), and those who had good CVH in both mid-life and late-life (Table 4). I do not think the authors reported any finding to support the statement that "improving CVH from mid- to late-life" is associated with lower risk of dementia. I suggest the authors re-work this Conclusions statement to accurately reflect their findings.

9. Introduction: After citing refs 15-19, which are cohort studies about cardiovascular health and risk of dementia, the authors assert that "the associations between patterns of CVH metrics from mid- to late-life and risk of dementia remain to be clarified." This sounds vague. What, specifically, remains to be clarified? I suggest the authors revise this sentence to be more specific about the gap in knowledge that remains after refs 15-19, which the present study will address.

10. Methods, Figure 1: The sample assessed for dementia in 1998 was 1,449 people. The sample in whom late-life CVH was measured in 1998 was 1,426 people. Why are these numbers different? What happened to the 23 people who were assessed for dementia in 1998 but who were not assessed for late-life CVH in 1998? Also, I suggest that the authors note more clearly in Figure 1 that mid-life CVH was assessed in 1972-1987 and that late-life CVH was assessed in 1998.

11. Methods: The authors calculated incidence rates of dementia as number of dementia cases divided by number of person-years of follow-up. This method may be OK, but does not account for the competing risk of death, and therefore the comparison of incidence rates across levels of CVH may be biased due to different rates of death across CVH levels. To overcome this limitation, the authors may want to consider estimating cumulative incidence of dementia from Cox proportional hazards models that account for death as a competing risk. See Andersen PK, et al. Int J Epidemiol. 2012;41:861-870. (In contrast, the hazard ratios for dementia are fine without accounting for the competing risk of death.)

12. Methods: The authors state that "the proportional-hazards assumption was verified using time-dependent covariates …" If I understand the authors' intent correctly, they added a timeXexposure interaction to the Cox model to allow for the exposure coefficient (hazard ratio) to vary over follow-up time. If so, the proper terminology is "time-dependent coefficient," not "time-dependent covariate." For an explanation of this distinction and of the appropriate method for adding a timeXexposure interaction to the Cox model, see Therneau T, et al. April 2, 2020. cran.r-project.org/web/packages/survival/vignettes/timedep.pdf.

13. Methods, Table S1: If the journal will allow it, I suggest incorporating Table S1 into the main manuscript instead of putting it in a supplement, as this table is critical for understanding the exposure measures. If the journal limits the main manuscript to a total of 5 tables + figures, then one possibility would be to merge Tables 2 and 3 into a single table to make room for one more table.

14. Results, Table 1: The statistical comparisons of characteristics of analytical sample #1 with characteristics of analytical sample #2, resulting in P values, are probably not strictly valid, because the two samples are not independent, but the methods for calculating the P values (t tests, chi-square tests) probably assume independent samples. If I understand correctly, the participants in analytical sample #2 are a subset of participants in analytical sample #1. The P values are not really meaningful or important, anyway, because the purpose of Table 1 is simply to describe the characteristics of the sample, not to make inference to a population. I suggest the authors omit P values from Table 1 and from the corresponding paragraph in the Results.

15. Results, Table 1: The percentage of participants in the intermediate category of smoking seems quite high, given that this category represents quitting smoking within the past 1 year, which is quite a narrow window given that smokers early in life may quit at any time over the course of decades. How did the authors determine whether former smokers had quit within the past 1 year?

16. Discussion, 3rd paragraph: The authors state that "We found no evidence for the association between late-life CVH metrics and the risk of dementia." While this may be because late-life CVH are less strongly associated with dementia, relative to mid-life (as the authors note), it may also be because of low precision due to small numbers of incident dementia cases in the analysis, relative to the estimated effect sizes (Table 3). The 95% CIs are very wide. I suggest the authors consider whether the precision is adequate in Table 3 for ruling out an effect of late-life CVH, and comment on this in the Discussion.

[LINK]

---

## [Decision Letter · Decision Letter 2]

28 Sep 2020

Dear Dr. Liang,

Thank you very much for submitting your revised manuscript "Cardiovascular health metrics from mid- to late-life and risk of dementia: a population-based study" (PMEDICINE-D-20-01344R2) for consideration at PLOS Medicine. 

Your revision was evaluated by a senior editor and discussed among all the editors here. It was also discussed with the academic editor, and sent to the original reviewers. The reviews are appended at the bottom of this email and any accompanying reviewer attachments can be seen via the link below:

[LINK]

In light of these reviews, I am afraid that still we will not be able to accept the manuscript for publication in the journal in its current form, but we would like to consider a revised version that addresses the reviewers' and editors' remaining comments. Obviously we cannot make any decision about publication until we have seen the newly revised manuscript and your response. 

We expect to receive your revised manuscript by Oct 05 2020 11:59PM. Please email us (plosmedicine@plos.org) if you have any questions or concerns.

We look forward to receiving your revised manuscript. 

Sincerely,

Thomas McBride, PhD 

Senior Editor 

PLOS Medicine

plosmedicine.org

1- Thank you for agreeing to make your data available. However, PLOS policy does not allow authors to be the primary contact for data access. Please provide a different contact for researchers applying for data access, e.g. a member of the CAIDE data management and maintenance committee who is not a study author.

2- Thank you for providing your pre-specified analysis plan and noting the changes or additions made. In the “Difference in analyses between planned and performed” section, please also note the reasons for the change (e.g. unexpected pattern in the data or in response to a reviewer request).

3- Title: please include the country “Cardiovascular health metrics from mid- to late-life and risk of dementia: a population based study *in Finland*”

4- Abstract Methods and Findings, second sentence: is it more accurate to say “We defined and scored *six of the seven* components of global CVH metrics…”?

5- Please provide p-values alongside 95% CIs throughout the manuscript.

6- Abstract Conclusions: Please address the study implications without overreaching what can be concluded from the data; the phrase "In this study, we observed ..." may be useful. Similarly for the first sentence of the Discussion Conclusions.

7- Was informed consent written or verbal? Please specify when mentioned in the Methods.

Comments from the reviewers:

Reviewer #1: We thank the authors for addressing the points raised in the previous review round, in particular the concern relating to competing risks. The additional Fine-Gray model results appear to generally affirm the previously-observed correlations regarding dementia outcomes between poor & intermediate/ideal CVH, for the main global CVH analysis.

However, the relatively low numbers of dementia cases, especially after stratification into subgroups as presented in Figure 2, has resulted in wide hazard ratio confidence intervals that leave some room for doubt, as well as seemingly counterintuitive findings such as intermediate/ideal biological CVH reported to have a HR of >7 compared to poor biological CVH. While possible explanations about biological CVH components such as blood pressure and total cholesterol have been suggested in the discussion, it would seem that the more fine-grained findings would indeed be well-served through larger-scale studies.

Nonetheless, the manuscript appears improved especially having taken into account suggestions from all reviewers, though there remain some minor issues:

1. Further details about the Fine-Gray competing risk model might be provided, possibly in supplementary material. In particular, how was the risk of death incorporated/quantified in the model?

2. The added statement of APOE being a genetic factor for dementia and being involved in lipid and lipoprotein metabolism (Page 12) might be supported by suitable references.

3. From S1 table, it appears that intermediate smoking (vs. poor) is also significantly associated with reduced risk of dementia (Page 14).

4. "having quitted more than one year" might be "having quitted for more than one year" (Page 9), and "Compared with people having the poor global CVH metrics in midlife" might be "Compared with people having poor global CVH metrics in midlife" (Page 13)

Reviewer #2: I appreciate the authors' thorough and responsive revision, including the authors' separation of global CVH metrics into behavioral and biological metrics and the addition of the S1 and S2 tables. I have only a few remaining minor comments:

1) In the introduction, the AHA's simple 7 are introduced, and, separately, the authors introduce that they will be evaluating CVH metrics. At the point of the introduction, it would be helpful to introduce that this paper will be evaluating the Life's Simple 7 factors, categorized as behavioral (sans diet - important to mention here) and as biological. With the recent revisions, the introduction of the Life's Simple 7 here doesn't quite tie into the rest of the paper because it isn't clear at the stage of the intro whether authors are going to evaluate the Life's Simple 7 verbatim or some other CVH metrics. Fine to refer to the Life's Simple 7 but make it clear that this is the background for the work, and what is being done in this paper, and how it's similar/different. This definition comes in the "Definition of CVH metrics" section later in more detail, but at the stage of the intro, I recommend you add a simple short sentence summarizing this exposure and how it's different/similar to Simple 7. 

Reviewer #3: The novelty of the study lies in the investigation of pattern of change in cardiovascular health from midlife to late life with incident dementia. However and despite the answers of the authors, to my view, the paper still suffers from the following major limitations:

- the analysis relies on 6 instead of 7 items of the cardiovascular health; the definition of the diabetic metric is incomplete

- there is too much attrition over time: 50% of the eligible population is lost from 1998 to 2005

- attrition can be handled with IPW technics but this was not done

- the change analysis (which is the novel aspect of the paper) is based on 47 incident dementia cases only 

- the results are hard to follow and to interpret, especially those on the change in cardiovascular health. 

Reviewer #4: In my opinion, the authors revisions in response to peer review comments—mine and other reviewers—resulted in an improved manuscript. I have no major comments and a few minor comments:

MINOR COMMENTS:

1. Table 2 (formerly Table 1) is now more informative then before about the nature of the sample. I still think the hypothesis test and P values are not very meaningful, given that the purpose of this table is simply to describe the characteristics of the sample, not to make inference to the population. The authors could consider simplifying Table 2 by omitting the P value columns.

2. The addition of cumulative incidence estimates accounting for competing risk of death to Tables 3 and 4 is informative. I suggest clarifying a few details about these estimates: (a) The length of follow-up for cumulative incidence should be reported, such as "20-year cumulative incidence," or whatever number of years of follow-up the estimates correspond to. (b) The word "rate" is not needed for cumulative incidence and could be omitted from the table column heading; the measure is "X-year cumulative incidence (%)," indicating the total incidence over the time span, not a rate. (c) Whether the cumulative incidence was adjusted for Model 1 or Model 2 covariates should be specified.

3. Figure 2 is interesting, but highlights clearly the lack of precision for estimating hazard ratios across so many different combinations of midlife and late life cardiovascular health levels (due to very small sample size for most groups). I know I suggested the authors undertake a more thorough analysis of the combination of midlife and late life cardiovascular health levels, and I am grateful they did so, but now I can see that the sample size is inadequate for such analysis to be very meaningful. The confidence intervals for most bars in Figure 2 are so wide that it is difficult to draw conclusions. Showing the confidence intervals visually instead of with numbers would enhance the figure, but would be difficult to show in the 3-dimensional layout. Thus, I suggest the authors consider revising Figure 2 into a 2-dimensional layout and show the height of the confidence intervals visually on the vertical axis rather than as printed numbers over the bars. Finally, the resolution of Figure 2 was very poor, such that the figure was nearly impossible to read. The authors should provide a high-resolution figure for publication.

4. Tables S1 and S2 are informative because they shed light on the heterogeneous influence of the 6 factors on risk for dementia. Table S2, in particular, helps us see one potential explanation for the late-life cardiovascular health score not being associated with lower dementia risk—"ideal" levels of late-life blood pressure, blood cholesterol, and body mass index are all associated with higher dementia risk, not lower risk. This is in line with prior evidence and especially helps to explain the findings illustrated in the bottom panel of Figure 2 (biological subscore). The authors addressed this briefly in the first paragraph of the Discussion; they could consider expanding this discussion a bit to highlight specific factors that behave differently in late life than in midlife for predicting dementia risk, and implications for using/interpreting the AHA cardiovascular health summary score for late-life data.

5. I agree with the authors' decision to omit the analysis of population attributable risk, for the reasons they gave in their response to Reviewer #1 comment (5).

--Evan L Thacker, PhD, Brigham Young University

[LINK]

---

## [Editor Report · Decision Letter 3]

27 Oct 2020

Dear Dr. Liang,

Thank you very much for re-submitting your manuscript "Cardiovascular health metrics from mid- to late-life and risk of dementia: a population-based study in Finland" (PMEDICINE-D-20-01344R3) for review by PLOS Medicine.

I have discussed the paper with my colleagues and the academic editor. I am pleased to say that provided the remaining editorial and production issues are dealt with we are planning to accept the paper for publication in the journal.

The remaining issues that need to be addressed are listed at the end of this email.

We look forward to receiving the revised manuscript by Nov 03 2020 11:59PM. 

Sincerely,

Thomas McBride, PhD

Senior Editor 

PLOS Medicine

plosmedicine.org

Requests from Editors:

1- Apologies for not asking last round, please add “cohort” to the study design. “… a population-based *cohort* study in Finland”.

2- In the Abstract Methods and Findings, please include the percentage of male participants alongside the mean age.

3- In the first sentence of the Abstract Conclusions, please edit to read: “... from midlife onwards is associated with a reduced risk of dementia *as compared with people with poor CVH metrics*.” or simlar.

4- Please add a similar phrase to point 5 of the Author summary, to make clear what the reduced risk of dementia is in comparison to.

5- Thank you for adding p-values throughout the text. In the tables, rather than a footnote to indicate when p< 0.05 or 0.01, please include the precise p-value in the table, down to p=0.001 (and p<0.001 for any smaller values).

6- Attrition could be added to the study limitations listed in the Abstract.

---

## [Editor Report · Decision Letter 4]

9 Nov 2020

Dear Dr. Liang, 

On behalf of my colleagues and the academic editor, Dr. Raquel C. Gardner, I am delighted to inform you that your manuscript entitled "Cardiovascular health metrics from mid- to late-life and risk of dementia: a population-based cohort study in Finland" (PMEDICINE-D-20-01344R4) has been accepted for publication in PLOS Medicine. 

PRODUCTION PROCESS

Before publication you will see the copyedited word document (within 5 business days) and a PDF proof shortly after that. The copyeditor will be in touch shortly before sending you the copyedited Word document. We will make some revisions at copyediting stage to conform to our general style, and for clarification. When you receive this version you should check and revise it very carefully, including figures, tables, references, and supporting information, because corrections at the next stage (proofs) will be strictly limited to (1) errors in author names or affiliations, (2) errors of scientific fact that would cause misunderstandings to readers, and (3) printer's (introduced) errors. Please return the copyedited file within 2 business days in order to ensure timely delivery of the PDF proof. 

If you are likely to be away when either this document or the proof is sent, please ensure we have contact information of a second person, as we will need you to respond quickly at each point. Given the disruptions resulting from the ongoing COVID-19 pandemic, there may be delays in the production process. We apologise in advance for any inconvenience caused and will do our best to minimize impact as far as possible.

EARLY VERSION

PRESS

PROFILE INFORMATION

Thank you again for submitting the manuscript to PLOS Medicine. We look forward to publishing it. 

Best wishes, 

Thomas McBride, PhD

Senior Editor 

PLOS Medicine

plosmedicine.org